# communications
# engineering

# Geodesy of irregular small bodies via neural density fields

Dario Izzo [1,2 ✉] & Pablo Gómez [1,2]

Asteroids' and comets' geodesy is a challenging yet important task for planetary science and spacecraft operations, such as ESA's Hera mission tasked to look at the aftermath of the recent NASA DART spacecraft's impact on Dimorphos. Here we present a machine learning approach based on so-called geodesyNets which learns accurate density models of irregular bodies using minimal prior information. geodesyNets are a three-dimensional, differentiable function representing the density of a target irregular body. We investigate six different bodies, including the asteroids Bennu, Eros, and Itokawa and the comet Churyumov-Gerasimenko, and validate on heterogeneous and homogeneous ground-truth density distributions. Induced gravitational accelerations and inferred body shape are accurate, resulting in a relative acceleration error of less than 1%, also close to the surface. With a shape model, geodesyNets can even learn heterogeneous density fields and thus provide insight into the body's internal structure. This adds a powerful tool to consolidated approaches like spherical harmonics, mascon models, and polyhedral gravity.

[1] Advanced Concepts Team, European Space Agency, European Space Research and Technology Centre (ESTEC), Keplerlaan 1, 2201 AZ Noordwijk, The Netherlands. [2] These authors contributed equally: Dario Izzo, Pablo Gómez. ✉email: dario.izzo@esa.int

The last two decades have witnessed the beginning of space exploration aimed at minor solar system bodies, such as asteroids and comets, beyond simpler fly-by missions. The mission NEAR visited the asteroid 253 Mathilde and was later successfully inserted into an initial orbit around 433 Eros in February 2000. From there, it thoroughly studied the asteroid and eventually performed a successful touchdown a year later as the mission ended[1]. After that, in 2005, the sample return mission Hayabusa, briefly touched down on the Muses Sea of 25143 Itokawa to collect samples from the asteroid surface and bring them back to Earth[2,3]. In 2011 the spacecraft Dawn surveyed Vesta for 12 months to then leave and reach, in 2015, its final destination, Ceres. In 2014, the Rosetta spacecraft and its Philae lander were able to visit and land on 67P Churyumov–Gerasimenko, examining at close proximity the activity of the frozen comet as it approached the Sun[4]. Hayabusa 2 sampled 162173 Ryugu[5] in 2018 and later returned the sample to Earth. Most recently, in 2020, OSIRIS-Rex obtained a sample from 101955 Bennu[6]. The continuation of this trend is clear as several missions are planned for this decade, such as Hera[7], ZhengHe[8], and Psyche[9]. Sampling and surveying asteroids and comets provide unique opportunities to study the history and development of the solar system[10–12]. Interest in visiting and surveying small solar system bodies was exclusively scientific up to recently, when several commercial entities showed interest into prospective asteroid mining and—with human space flight ambitions once again looking beyond Earth orbit—since topics surrounding in-situ resource utilization on minor planets are now of particular interest[13–15].

In these types of interplanetary missions, knowledge of the geodesy of the investigated bodies plays a critical role in successfully performing orbital and surface proximity operations, in closely tracking touch-and-go trajectories as well as in evaluating the collected measurements and observations. The gravity field generated by the body and the acceleration induced on the spacecraft allows the precise planning and execution of mission operations[16,17], while knowledge of the body shape and of its internal mass distribution—which may give insight into the body's origin and composition—are of interest to both scientists and mission operators[18–20]. The gravitational field of a celestial body is, for most operational purposes, typically represented by a spherical harmonics expansion of the gravitational potential with coefficients learned via Kalman filtering techniques[16,17]. Unfortunately, this approach loses its appeal as the body irregularities become more important[21,22]. Other options, such as mascon models[23,24] and polyhedral gravity[25,26] can overcome some of these difficulties, but also introduce other requirements, such as the need for a shape model or the assumption of a homogeneous internal density. In the case of ESA's Hera mission[7], planned to rendezvous with the 65803 Didymos-Dimorphos system in late 2026, it is foreseen to operate the spacecraft at different distances from the bodies during an early characterization phase followed by a second detailed characterization phase. In the first phase, the mass and a preliminary shape model of the bodies will be assembled and later used, during the second phase, to obtain insights into the surface and internal properties. During these phases, the bodies' gravity fields will be measured and important data produced that will be used later during accurate geodetic studies.

In preparation for this type of work, we introduce geodesyNets, a new, generic and unified approach to gravity representation able to reach competitive results with respect to the state-of-the-art. To develop geodesyNets, we took inspiration from recent trends and breakthroughs in artificial intelligence and computer vision, such as the Generative Query Networks by ref. [27] or the Neural Radiance Fields by refs. [28–30] who introduced novel neural

network architectures and training methods for three-dimensional scene reconstruction from two-dimensional images. In a more abstract sense, their works follow a general, emerging trend of utilizing neural networks as a method for solving inverse problems[31–34]. These works showed how deep networks can represent complex three-dimensional scenes with great accuracy when appropriately trained. Thus, if a training procedure can be found that makes use of gravitational measurements, it is only consequential that a deep network could learn also the complex, irregular, mass geometry of Solar System bodies.

With geodesyNets, we solve the body-specific problem of gravity inversion and shape reconstruction using a neural network as an approximator for the distribution of mass in an enclosing cubic volume. The investigated bodies in our test cases are asteroids or comets. We are able to reach competitive accuracy compared to previous approaches with notably fewer assumptions about body shape and density distribution. The trained geodesyNets are able to simultaneously represent both body shape and density distribution accurately. Further, the approach is able to incorporate body shape information for improved results. Assuming the availability of a shape model, we demonstrate the successful application of our technique to bodies with a heterogeneous density distribution. Finally, we show that training geodesyNets is a computationally efficient process, which bears the promise of on-board applicability. To allow for easy replication of our results, we provide all code (https://github.com/darioizzo/geodesynets) and data online (https://zenodo.org/record/4749715#.YJrR6OhfiUk).

## Results

**Geodesy artificial neural networks: geodesyNets**. We represent and study the geodetic properties of generic celestial bodies using fully connected neural networks that we call geodesyNets. A geodesyNet represents the body density directly as a function of Cartesian coordinates. In other words, the parameters of the network (i.e. the weights and biases) represent the density as a differentiable, and thus continuous, field. ref. [28] introduced a related network architecture called Neural Radiance Fields (NeRF) to represent three-dimensional objects and used it to reconstruct complex scenes with impressive accuracy learning from a set of two-dimensional images. The training of a NeRF solves the inverse problem of image rendering as it back-propagates the difference between images rendered from the network and a sparse set of observed images. Similarly, the training of a geodesyNet solves the gravity inversion problem. The geodesyNet network learns from a dataset of measured gravitational accelerations back-propagating the difference to the corresponding accelerations computed from the density represented by the network. The similitude between our technique and previous works on an implicit neural representation that inspired us[27,28,35] cannot, unfortunately, be brought any further since we address here a fundamentally different inversion problem, that of a gravitational field, hence the underlying physics and resulting equations and implemented methods diverge substantially.

At the end of its training, our network provides a differentiable (and thus continuous) expression mapping the position within a cubic volume $V$ to a body density $\rho(x, y, z)$ compatible with the observed accelerations. Analogously to the radiance fields of the NeRF, we refer to $\rho$ as a neural density field. The neural density field can be used to study the body's internal structure, to compute the gravitational potential field and accelerations outside and inside the body, as well as to derive quantities such as the spherical harmonics coefficients that depend uniquely on said density function. Because of the mathematical properties of artificial neural networks as universal approximators, recently

re-discussed in depth by ref. [36], sharp density discontinuities and sudden variations of the body structure can also be represented, and, in fact, a geodesyNet implicitly learns a plausible representation of the asteroid surface as a two-dimensional discontinuity embedded in the three-dimensional space. In other words, indicating with $V_B$ the volume actually occupied by the body, a geodesyNet learns to infer a vanishing density outside $V_B$ and to jump to a finite value when approaching the body surface $\partial V_B$, without ever being trained on where $\partial V_B$ actually is.

The overall architecture of a geodesyNet is shown in Fig. 1 (more details are given in the method section). First, the Cartesian coordinates $x$, $y$, $z$—indicating the position of a point within the hypercube $V$—are fed into an encoding layer mapping them to a representation suitable for the network. Rahaman et al.[37] have recently noted how the expressivity property of deep networks, able to fit even random input-output mappings, comes with a spectral bias manifesting in the network learning low-frequencies first. This is important to the geodetic application proposed here as an irregular body shape, to start with, might suffer from a poor representation of important high-frequency contents, for example, at the surface $\partial V_B$. The encoding layer allows experimenting with transformations of the Cartesian coordinates able to control the network's spectral bias. Eventually, we find that a direct Cartesian encoding, coupled with periodic activation functions between layers[35], offers optimal performance in terms of the resulting quality of the neural density field (supplementary methods, Table S5). After the encoding layer, a number of fully connected layers follow, forming the main body of the network with its learnable parameters. The network output, i.e., the neural density field indicated with $\rho$, is then used to compute the gravitational acceleration at the measurement point. A detailed account of how this is achieved via a numerical quadrature is reported in supplementary methods, Supplementary Method 2. The numerical quadrature necessitates the evaluation of the network at $N$ distinct points inside the hypercube $V$ and its numerical precision will depend on it, as shown in supplementary methods, Fig. S1. The difference between the computed and the measured (ground-truth) values is expressed in a loss function $\mathcal{L}(\theta)$ minimized as to update the model parameters $\theta$.

The result is a process that gradually learns a three-dimensional model of the body density compatible with the measured gravitational field, as seen, for example, in Fig. 2, where the learning process is shown in the case of synthetic gravitational data generated for the comet Churyumov–Gerasimenko. The learned model can be used to determine the geometric shape and (to some extent) the internal structure of the body, its orientation in space, and its gravity field, thus allowing the geodetic properties of the body to be fully determined from the parameters $\theta$ defining the neural architecture. The same pipeline can also be applied to measurements of the gravitational potential, resulting in a neural density field with similar accuracy.

**The ground truths**. To produce synthetic values for the measurements $y_i$ of gravitational accelerations, we use mascon models

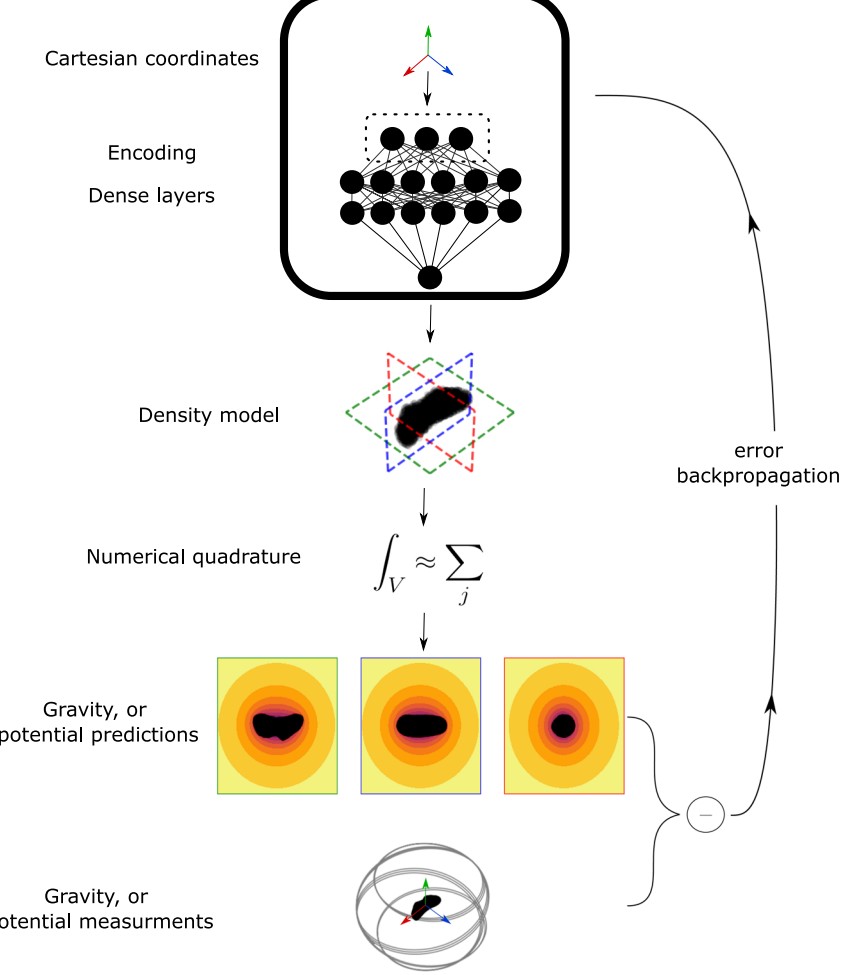

**Fig. 1 GeodesyNets.** Overall schematics of the process of training a geodesyNet.

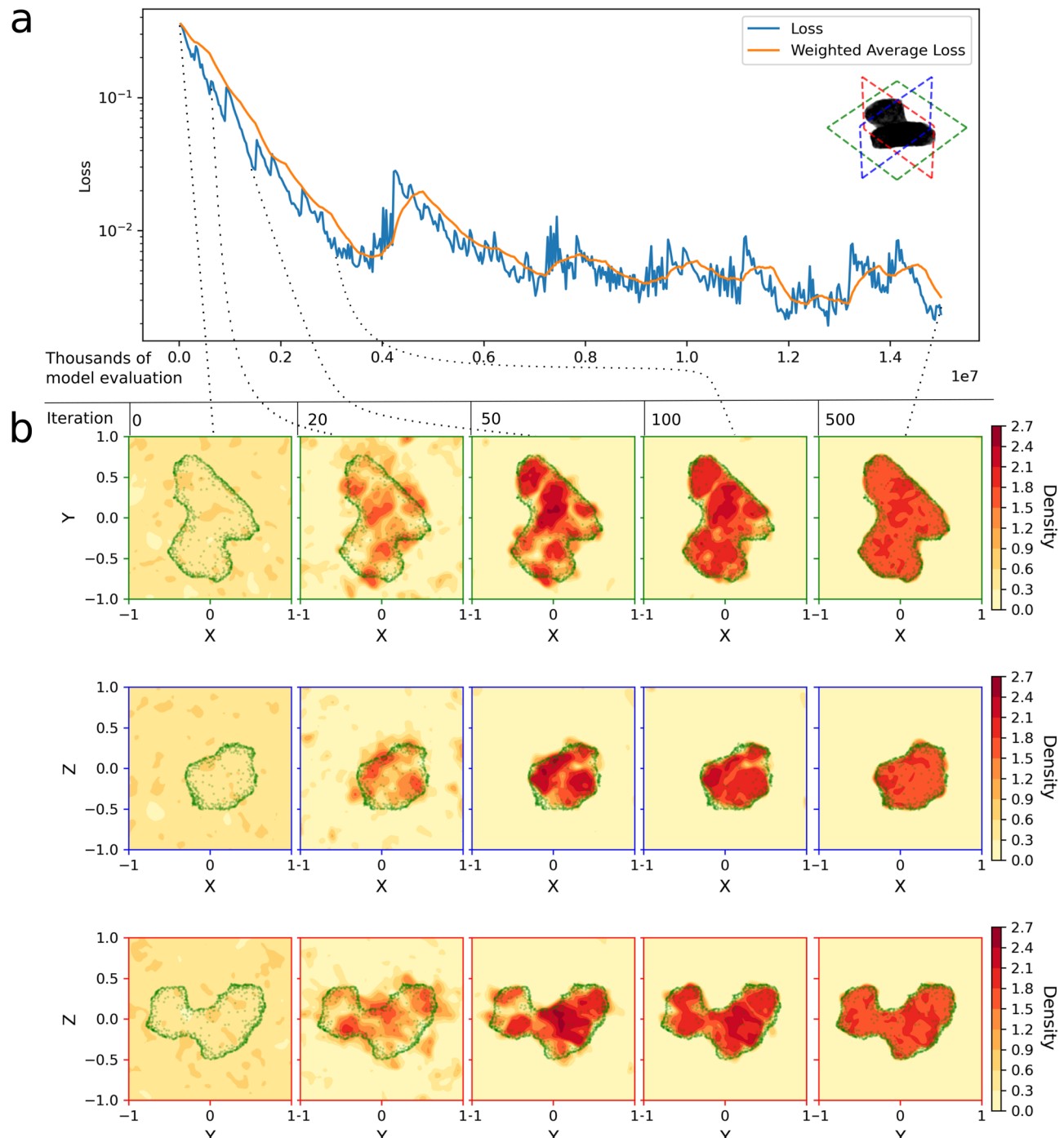

**Fig. 2 Exemplary training of a geodesyNet on the gravitational acceleration of a homogeneous density Churyumov–Gerasimenko comet model (visualized in green).** Non-dimensional units are used (see supplementary methods, Table S3). The training loss is shown in (**a**), while the learned mass distribution in the hypercube $V = [−1, 1]^3$ is shown in (**b**) for the $xy$, $xz$, and $yz$ cross sections. As learning progresses (from left to right), the mass distribution evolves to reconstruct the correct asteroid shape and a uniform internal density.

representing mass distributions with varying degrees of heterogeneity. When enough mascons are considered, this discontinuous representation approximates well a continuous distribution, while also offering the flexibility of easy and direct implementation. Each mascon model is a list of tuples $\mathcal{M} = \{(x_i, y_i, z_i, m_i) \quad i = 1..n\}$ and allows to compute the ground-truth gravitational acceleration at a generic point $\mathbf{r}_i$ via the formula:

$$\mathbf{a}(\mathbf{r}_i) = -G \sum_{j=1}^{n} \frac{m_j}{r_{ij}^3} \mathbf{r}_{ij},$$

where $G$ is the Cavendish constant, and $m_j$ is the generic mascon mass placed at $\mathbf{r}_j$. We consider the asteroids 433 Eros, 25,143 Itokawa, and 10,1955 Bennu and the comet 67P Churyumov–Gerasimenko, as well as a fictitious Planetesimal and a toroidal-shaped body we call Torus. The specific test cases were chosen to compare to prior research or study difficult cases. First, we generate a mascon model for each body that represents a homogeneous mass distribution. Then, in the case of Bennu, Itokawa, and Planetesimal, we generate additional mascon models representing a heterogeneous mass distribution. For Bennu, we

**Table 1 Learned models—mean absolute and relative acceleration errors at three altitudes (low, medium, and high).**

| | Body | Sampling altitudes | | | Absolute errors | | | Relative errors | | |
|---|---|---|---|---|---|---|---|---|---|---|
| | | $h_{low}$[m] | $h_{med}$[m] | $h_{hi}$[m] | $\epsilon_{low}$[m/s$^2$] | $\epsilon_{med}$[m/s$^2$] | $\epsilon_{hi}$[m/s$^2$] | $\epsilon_{low}$[%] | $\epsilon_{med}$[%] | $\epsilon_{hi}$[%] |
| HMG | Bennu | 14.1 | 28.2 | 70.4 | 2.63e-08 | 4.75e-09 | 6.89e-10 | 0.11 | 0.02 | 0.005 |
| | Churyumov–Gerasimenko | 125 | 250 | 625 | 1.13e-07 | 2.02e-08 | 2.20e-09 | 0.19 | 0.04 | 0.006 |
| | Eros | 817 | 1630 | 4080 | 2.24e-06 | 4.45e-07 | 5.52e-08 | 0.16 | 0.04 | 0.01 |
| | Itokawa | 14 | 28 | 70.1 | 3.15e-08 | 6.35e-09 | 1.06e-09 | 0.15 | 0.04 | 0.01 |
| | Planetesimal | 125 | 250 | 625 | 5.69e-08 | 1.31e-08 | 3.43e-09 | 0.11 | 0.03 | 0.011 |
| | Torus | 125 | 250 | 625 | 1.41e-07 | 3.74e-08 | 8.49e-09 | 0.28 | 0.09 | 0.034 |
| HTG | Bennu | 14.1 | 28.2 | 70.4 | 4.70e-08 | 9.57e-09 | 1.57e-09 | 0.20 | 0.05 | 0.011 |
| | Itokawa | 14 | 28 | 70.1 | 4.27e-08 | 9.36e-09 | 9.33e-10 | 0.20 | 0.05 | 0.009 |
| | Planetesimal | 125 | 250 | 625 | 9.90e-08 | 2.53e-08 | 4.22e-09 | 0.20 | 0.06 | 0.014 |

Altitudes for validation are chosen depending on the body size as a fraction of its diameter.
HMG homogeneous, HTG heterogeneous.

introduce an equatorial region with lower density with a reduction factor $f = 2$. For Itokawa, we make the asteroid head heavier, augmenting its density by a factor $f = 1.6$. For Planetesimal, instead, we create an internal cavity of spherical shape, thus radically changing the body topology. Overall, the ground truths were chosen to represent real objects and, additionally, objects that can explore potential edge cases. In the case of Planetesimal, a cave inside the object is modeled and the Torus serves to showcase a highly non-convex object.

The resulting mascon models are visualized in the supplementary methods, Fig. S2, and their parameters are reported in the supplementary methods, Table S3. More details on the generation process are also given in the supplementary methods, Supplementary Method 4. Note that the body sizes of the chosen bodies vary greatly as, e.g., Bennu has a diameter of merely 525 m while Eros has one of almost 17 km. For convenience and consistency, we, therefore, introduce and use non-dimensional units for all the obtained models. As a result, the integration volume $V$ for all cases is reduced to be the cube $[-1, 1]^3$ which is ensured to contain all of the asteroid mascons.

**Learned models**. We apply the training pipeline depicted in Fig. 1 to obtain a neural density field for each of the nine mascon models generated, including the heterogeneous models for Bennu, Itokawa, and Planetesimal. The quality of the final result is assessed quantitatively by comparing the ground-truth acceleration from the mascon model to the accelerations caused by the neural density field at 10,000 random validation points located at specific altitudes. For each body, we fix a low, medium, and high altitude corresponding respectively to 0.05, 0.1, and 0.25 units of length. An example of the resulting validation points in the case of Eros is shown in the supplementary methods, Fig. S3 for the medium-altitude case. Additional details on the training, validation, and sampling are given in the Methods section. In Table 1, we show the results in terms of mean absolute error and mean relative error. For all cases considered, our geodesyNet is able to learn the mass density in the volume $V_B$ such that it reproduces the ground-truth gravity field with a relative error between 0.11 and 0.28% on all bodies at all tested altitudes.

A visual indication of the achieved accuracy for all the homogeneous solar system bodies is given in Fig. 3, where the neural density field is plotted against the mascon ground-truth. Note that even small-scale surface features, such as larger rocks and craters, are reconstructed to some extent. For completeness, we also report the relative error close to the surface, in the case of the heterogeneous Itokawa model, in Fig. 4. The error distribution across the asteroid surface appears overall to be quite uniform along the asteroid body, revealing how the neural density field is able to balance errors caused by the presence of complex surface features. It is worth to note how most of the relative error is, as expected in the case of the heterogeneous Itokawa, concentrated around the heavier asteroid head where the close proximity to the larger source of the gravitational field is penalizing.

**Comparison with existing methods**. The use of machine learning to represent the gravity field of small bodies has been the subject of two recent works[38,39]. The work of ref.[39] proposes the use of a Hopfield network to represent and learn on-board the spherical harmonic coefficients. Unlike GeodesyNets, such a representation, useful for the use of preliminary navigational models, is subject to the same convergence concerns as any model based on spherical harmonics. The original work of ref. [38], instead, uses a network to represent directly the gravity potential. This approach requires a pre-existing model of the gravity field to learn from and must inform the loss as to enforce the Laplace equation. Since a GeodesyNet represents the density $\rho$ directly, such an equation is satisfied by construction and needs not to be enforced.

A direct quantitative comparison with prior literature is difficult as a common validation practice for modeling irregular gravity fields has not been established yet and many of the available modeling approaches either rely on different assumptions or make use of data not made available, and are thus essentially not reproducible. It is nevertheless of interest to look to published and independent works, with the understanding that exact comparisons are not possible at this stage. In this respect, a good candidate to establish a benchmark is perhaps the work by ref. [24], who provides a detailed analysis on the performance of a state-of-the-art method representing the gravity field of a perfectly uniform Eros, albeit informed by a shape model. Their work makes use of a hybrid description of the asteroid gravity field, hybridizing a mascon model to a spherical harmonics model. A detailed comparison to their work, provided in supplementary methods (see Supplementary Method 5), reveals how the GeodesyNet approach is able to reach similar accuracies, remarkably even if no shape model is assumed.

In order to allow for a rigorous quantitative comparison and introduce a further solid benchmark, we implemented our own version of a pure mascon approach, one that does not rely on any shape information, and we used it to create a gravitational representation for all the homogeneous bodies here investigated. We refer to this new model as a masconCUBE: a uniform grid of $N$ masses $m_j$ placed in a regular grid inside the volume $V$. Such a model does not make use of a shape model and is thus an appropriate benchmark to study the representation quality of

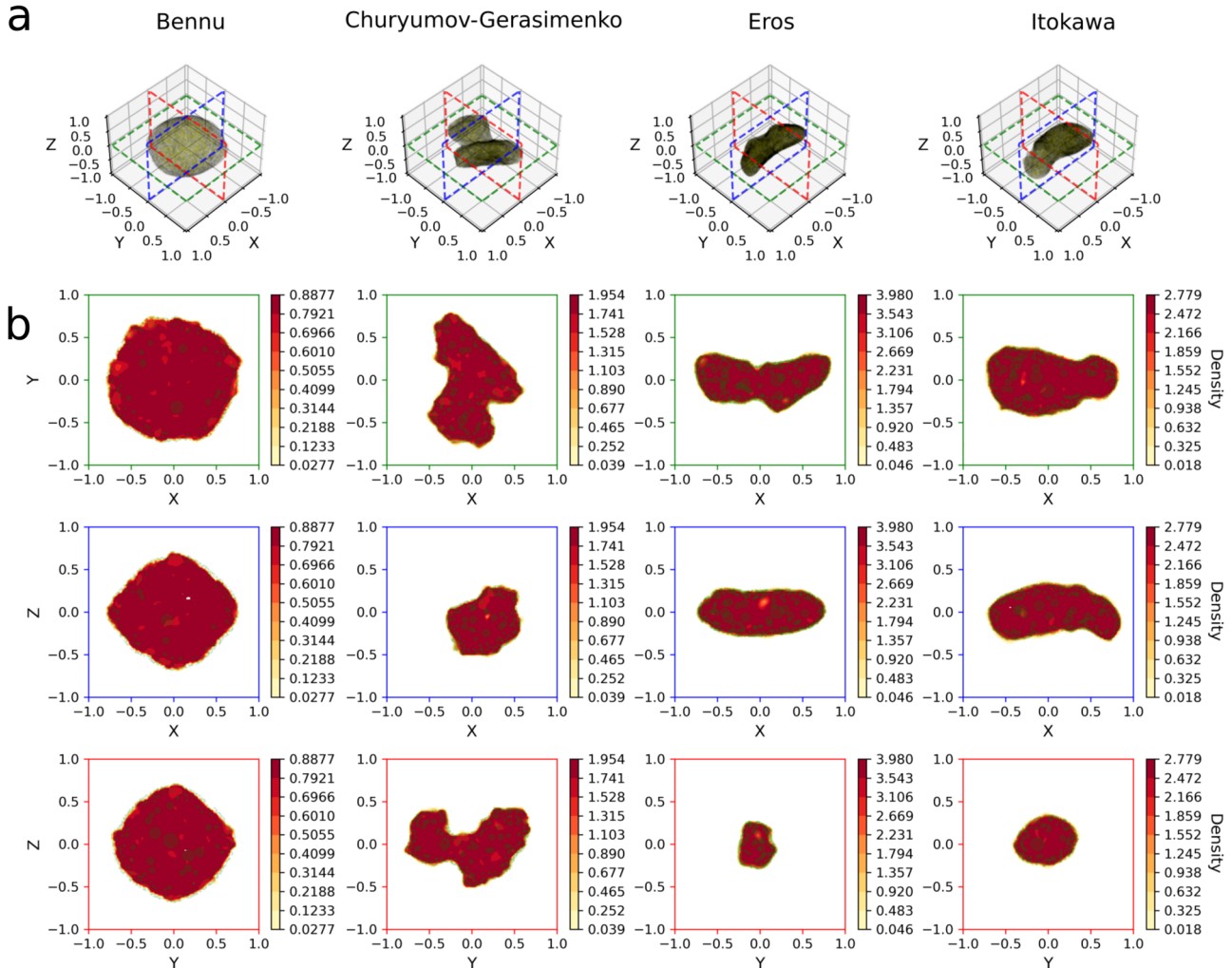

**Fig. 3 Standard training—results on the four (homogeneous) solar system bodies.** Ground-truth models are in (**a**), as well as the selected slice location for the slices depicted below. For each slice, a heatmap of the neural density field is overlaid with the mascons (green) in (**b**). Only mascons within a small distance to the selected slice are shown, hence the apparent mascon sparsity. The neural density fields are able to correctly reconstruct the body shape as well as their internal homogeneous density.

GeodesyNets. For each of our homogeneous bodies, we thus train a masconCUBE (i.e., we find the values of all the $N$ masses $m_j$) on the very same gravitational data we used for the corresponding GeodesyNet. In past works, large mascon models have been trained using the batch least square estimation methodology[24,40]. For the purpose of the comparison we use, instead, the same procedure used to train the GeodesyNet weights. Two small changes are necessary, namely a readjustment on the learning rate, set to 0.1, and on the definitions of the model parameters $\eta_j$ which are defined as to satisfy $m_j = \eta_j^2$ as to ensure positive values for all mascon masses as well as improved gradient information. We make sure that the number of mascon $N$ is the same as the number of parameters (weights and biases) used in our largest networks and, in particular, $N = n^3 = 91,125$ corresponding to $n = 45$ mascons per cube side. The results are summarized in Table 2, where the relative error on the acceleration vector is reported averaged over points randomly sampled close to the surface (i.e., within ≈0.15 length units from the surface) and further away (i.e., between ≈0.15 and ≈0.3 length units from the surface) The novel GeodesyNet model describes the gravitational field of the irregular bodies considered with great accuracy, in particular, close to the body surface. A comparison in terms of the spherical harmonics computed using both methods

is also given in the supplementary methods (see Supplementary Methods 6, 7, and Table S6).

**Taking advantage of a shape model.** If additional information on the body shape $\partial V_B$ is available, this can be seamlessly integrated into the geodesyNet training. A geodesyNet can, in fact, be trained to represent, instead of the body density, the density variation from a homogeneously distributed density inside $\partial V_B$. We refer to this variant of the training pipeline as differential training (see more details in supplementary methods, Supplementary Method 3). We apply the differential training pipeline only to the three heterogeneous cases, as for the homogeneous bodies, the neural density field learned by the differential approach is trivial as it would vanish entirely, and so would the obtained relative errors. Figure 5 visualizes the heterogeneous ground truths and the network's predicted distributions. In Table 3, the relative and absolute errors on the predicted accelerations are given. In comparison to the previously trained geodesyNets (see Table 1), differential training achieves similar results in terms of acceleration error, in particular, halving the relative error at low altitudes. The advantage of differential training becomes clear in the qualitative description of the

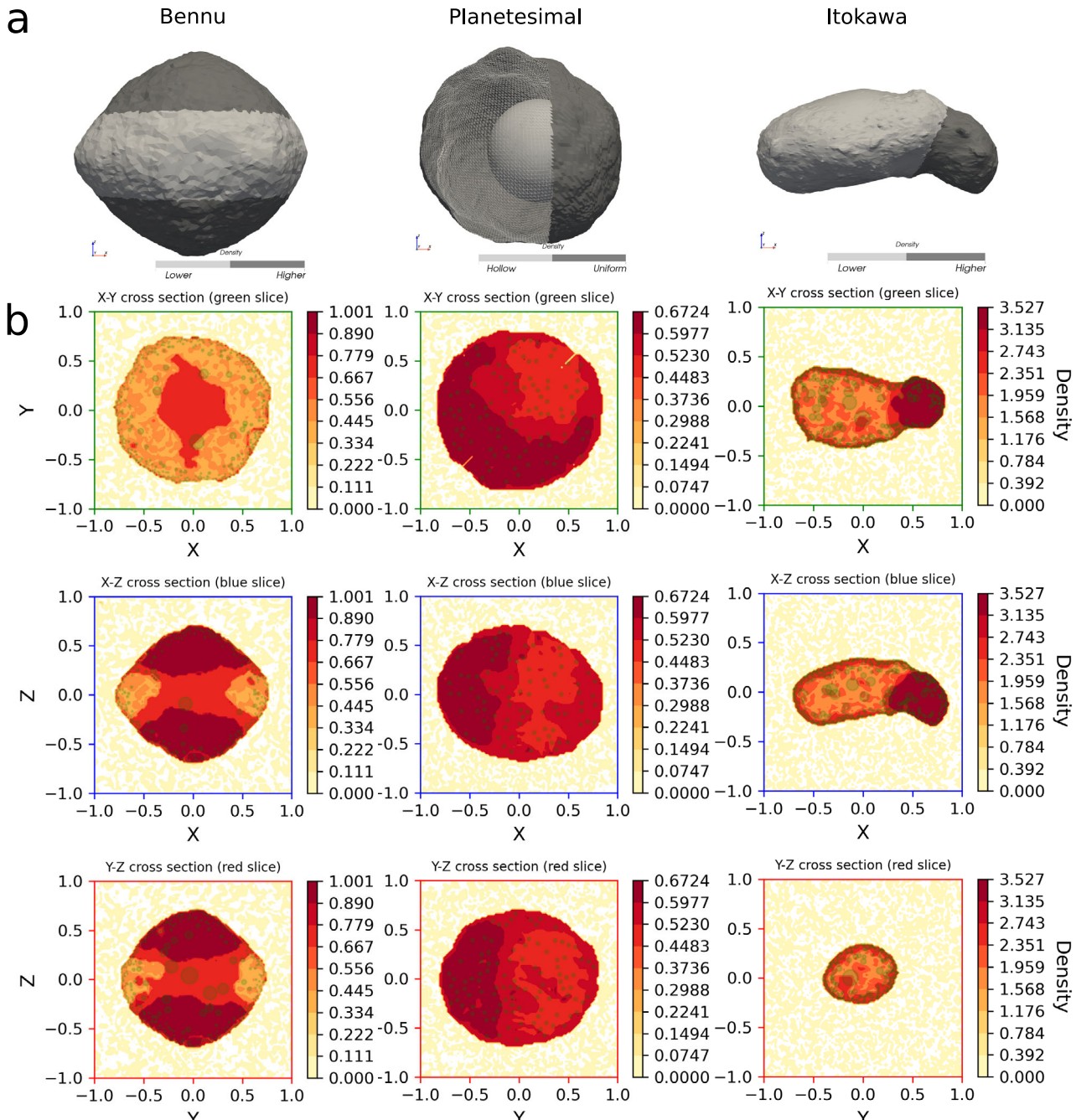

**Fig. 4 Differential training—results on the three heterogeneous bodies.** Ground-truth modelsare shown in (**a**) indicating the lower and higher density regions by shade. For each slice, a heatmap of the reconstructed neural density field is overlaid with the mascons (green) in (**b**). Only mascons within a small distance to the selected slice are shown, hence the apparent mascon sparsity.

**Table 2 Relative errors [%] using GeodesyNets and a comparable mascon representation.**

|  |  | Bennu | Churyumov–Gerasimenko | Eros | Itokawa |
|---|---|---|---|---|---|
| GeodesyNet | low | **0.72** | **2.30** | 1.82 | 2.13 |
|  | hi | 0.02 | **1.75** | 0.17 | **0.38** |
| masconCUBE | low | 1.00 | 2.87 | 2.39 | 2.47 |
|  | hi | **0.01** | 2.03 | **0.13** | 0.70 |

Close to the surface, the advantage offered by the use of GeodesyNet is revealed (bold numbers indicate the best model).

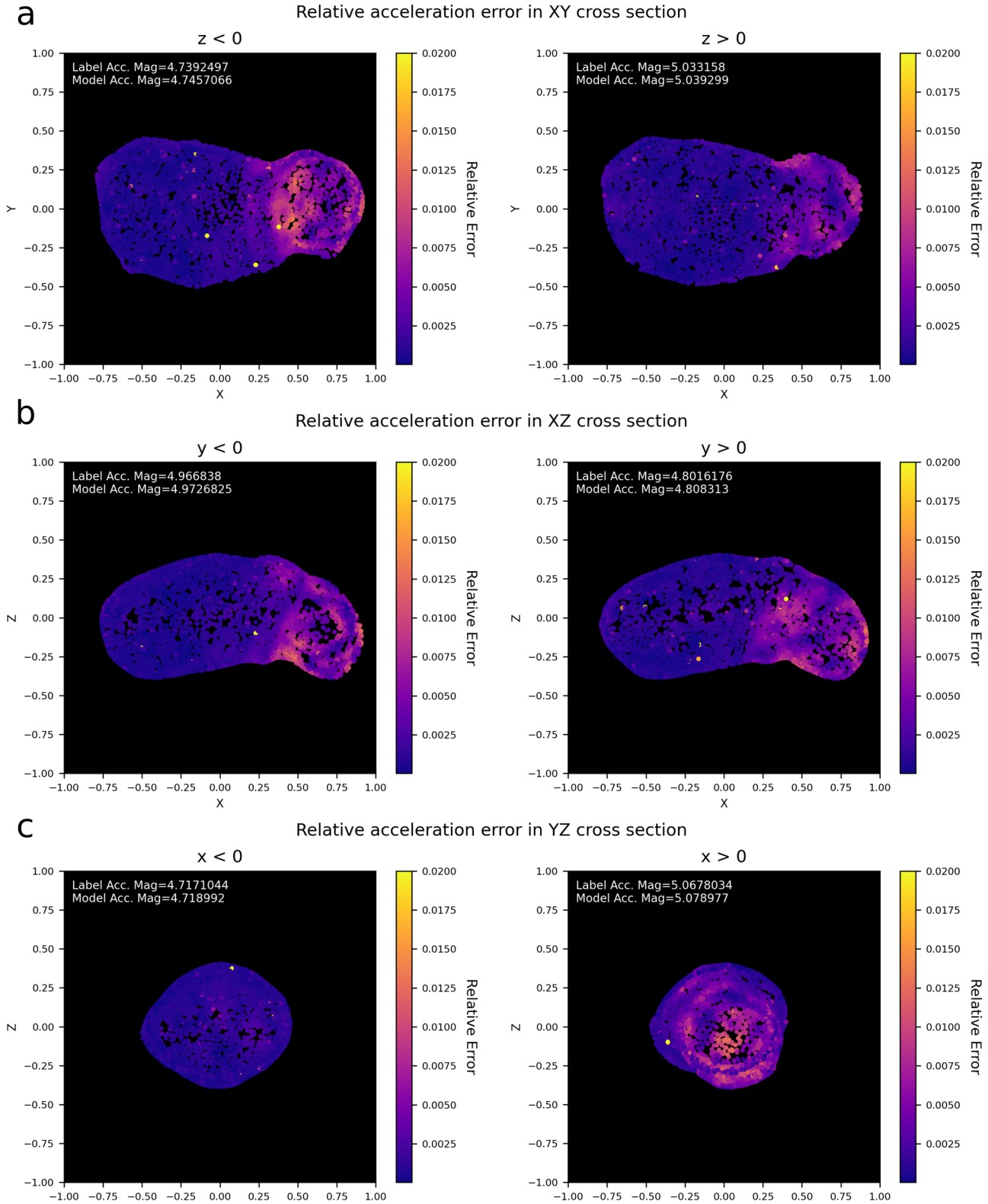

**Fig. 5 Relative acceleration errors.** Visualization of relative errors of the gravitational acceleration predicted by the trained geodesyNet close to the body's surface. Different hemispheres are displayed in (**a–c**). Points are located 17.5 m above the asteroid surface (0.05 in non-dimensional units). The spread is quite uniform, except for points closer to the heavier asteroid head that are correlated to higher errors, as expected.

resulting density field resulting closer to the actual heterogeneous density distribution.

In the heterogeneous Bennu case, the neural density field clearly shows the presence of a lower density region at the equator

(see Fig. 5 left column) and it also outputs density values compatible with the ratio $f = 2$ used to create the ground-truth density difference between polar and equatorial regions. Note that the asteroid core is left at a slightly higher density than the outer

**Table 3 Differential training—mean absolute and relative acceleration errors for heterogeneous bodies at three altitudes (low, medium, and high).**

| Heterogeneous body | Sampling altitudes | | | Absolute errors | | | Relative errors | | |
|---|---|---|---|---|---|---|---|---|---|
| | $h_{low}$[m] | $h_{med}$[m] | $h_{hi}$[m] | $\epsilon_{low}$[m/s²] | $\epsilon_{med}$[m/s²] | $\epsilon_{hi}$[m/s²] | $\epsilon_{low}$[%] | $\epsilon_{med}$[%] | $\epsilon_{hi}$[%] |
| Bennu | 14.1 | 28.2 | 70.4 | 4.07e-08 | 1.19e-08 | 7.67e-09 | 0.10 | 0.03 | 0.031 |
| Itokawa | 14 | 28 | 70.1 | 2.49e-08 | 1.45e-08 | 1.01e-08 | 0.12 | 0.08 | 0.091 |
| Planetesimal | 125 | 250 | 625 | 3.55e-08 | 2.29e-08 | 1.84e-08 | 0.08 | 0.06 | 0.071 |

Latitudes for validation are chosen depending on the body size as a fraction of its diameter.

**Table 4 Comparative table of different fundamental approaches able to represent and learn gravity fields of three-dimensional bodies.**

| | Approach | | | |
|---|---|---|---|---|
| | Masc. | Harm. | Poly. | geodesyNets |
| Differentiable | ✗ | ✓ | ✓ | ✓ |
| Inside Brillouin sphere | ✓ | ✗ | ✓ | ✓ |
| Heterogeneous densities | ✓ | ✓ | ✗ * | ✓ |
| Shape model not needed | ✓ | ✓ | ✗ | ✓ |
| Can utilize shape model | ✓ | ✗ | ✓ | ✓ |
| Accurate in the near field | ✗ | ✓ | ✓ | ✓ |

Mascons (Masc), Spherical Harmonics (Harm), and Polyhedral gravity (Poly) models are compared, qualitatively, to our GeodesyNets.
*Note that models based on Polyhedral gravity can be used to some degree for heterogeneous densities but such a practice is impractical and limited.

layers at the equator—a solution selected by the network bias among the ones compatible with the observations. For the heterogeneous version of Itokawa, the trained geodesyNet is able to precisely reconstruct the higher density region at the asteroid's "head"—closely matching the ground-truth (see Fig. 5 right); also from a quantitative point of view as the predicted density values are compatible with the correct ratio $f = 1.6$. A small inaccuracy is notable in the predicted slightly higher density of a thin layer close to the surface, likely a numerical artifact caused by the mismatch between the shape of the asteroid and the mascon approximation near the surface. Mascon models suffer from inaccuracies next to the body surface where their point mass nature becomes evident. For the case of Planetesimal, while differential training improves the description of the surrounding gravity field, reducing the relative error from 0.2 to 0.08% in the near field—see Tables 3 and 1—we note that the obtained neural density field fails to fully reproduce closely the cavity inside the asteroid, settling instead for a solution where the ratio between the density inside the cavity and the one outside is not zero. We must note that the gravity inversion problem is, for this particular shape, particularly challenging as the heterogeneity chosen for this rather symmetrical Planetesimal nears a configuration where the Shell theorem would apply, exacerbating the ill-posed nature of the gravity inversion problem[41–44].

## Discussion

**Gravity representation.** Prior to our work, three main representations of a gravity field have been widely studied and used in the context of geodesy: spherical harmonics, mascon models, and polyhedral gravity representations[25]. A qualitative comparison of these approaches is displayed in Table 4.

The spherical (or the spheroidal) harmonics approach allows the representation of a generic gravity field via the coefficients of its Fourier series expansion in spherical (or similar) coordinates. Its use is particularly suited for bodies, such as large planets and moons, that have strong axial symmetry and regularity. For example, thanks to accurate data collected during missions such as the Steady-state Ocean Circulation Explorer (GOCE)[45] or the Gravity Recovery and Climate Experiment (GRACE)[46], spherical harmonics expansions could be computed to describe the Earth's gravity field up to wavelengths of the order of ≈160 km [47], including terms of degree up to 250. However, applying the same technique to achieve comparable precision for irregularly shaped bodies is possible but troublesome. Inside the Brillouin sphere[48], the convergence of spherical harmonics expansions is known to be erratic[21] and requires special attention[22]. Outside the Brillouin sphere, the convergence becomes slower as soon as the body shape has irregularities departing notably from a simple reference triaxial ellipsoid. In the case of small solar system bodies such as comets and asteroids, the use of such an expansion outside the Brillouin sphere, when possible, requires an increasingly high number of coefficients. For example, ref. [22] developed a spheroidal harmonics model for the relatively regularly shaped asteroid Bennu using terms of degree up to 360 and reported relative errors, on the obtained potential, of the order of ≈1% in the exterior field (e.g., at 5–20 m from the Brillouin sphere). It has been proposed to make use of a different expansion for the interior gravity[49] at the cost of introducing an artificial boundary at the Brillouin sphere surface. A geodesyNet model makes no distinction between the inner and exterior fields as it equally learns to match gravitational observations inside and outside of the Brillouin sphere. Its neural architecture, based on the neural radiance fields, is not impacted as much by the complexity of the represented bodies, as shown also by the work of ref. [28] that reports success in encoding many different three-dimensional scenes using a fixed amount of parameters.

A second method classically used to represent the gravitational field is that of filling the volume occupied by the body with point masses or "mascons". Each mascon is assigned a mass so that the total body mass is reconstructed. All our ground-truth gravity fields were generated this way. The approach has the great advantage of its simplicity and it is straightforward to implement, but it also has several deficiencies when employed to represent

observed gravity fields. The number of mascons needed to achieve accuracies comparable to that of spherical harmonics in the description of the exterior field is very large[23]. In the interior field, and specifically close to the surface, the accuracy is troublesome as it becomes unclear whether a point is underneath or just over the body surface. These drawbacks can be reduced using hybridized techniques[24] and introducing knowledge of the shape and composition of the asteroid. A geodesyNet, as shown in the result section, is able to achieve comparable performances to that of mascon hybridized techniques while using no prior information on the body shape and returning a continuously differentiable representation of the internal body density.

The last approach, polyhedral gravity, has been widely used in the context of irregular bodies' gravity. Therein, the divergence theorem, or Gauss's theorem, is used to transform the triple volume integral needed to compute the gravitational acceleration or potential into the integral along the asteroid surface of its flux[23,25]. The surface is then approximated by a polyhedron and an analytical formula is derived that enables computing the gravity produced by any polyhedral body having a homogeneous density. This technique is only valid for homogeneous bodies and requires the availability of a high-fidelity model of the body shape to derive a polyhedral model. Often, this is computed on-ground thanks to reconstruction techniques based on the images available from on-board cameras. This method can also have a high computational complexity as a large amount of evaluations of functions like arctangents and logarithms is needed[26]. A geodesyNet, on the other hand, does not need a homogeneous density body to be able to learn a representation of the gravity field, but it is able to use, if available, the shape information to improve the quality of its predictions and to build plausible models for the internal structure of heterogeneous asteroids. In Table 4, we have summarized the different properties of fundamental representations of a gravity field, highlighting the broad applicability and various use cases of the proposed geodesyNets.

**Gravity inversion**. From a methodological viewpoint, there are several distinguishing factors and relevant parallels of geodesyNets in comparison to previous approaches. One noteworthy aspect of geodesyNets is the ability to serve several purposes at once: after training, the same geodesyNet can be used to represent both the gravitational field around a body as well its shape and density. In detail, the geodesyNet learns a representation for the gravitational field outside of the body, for the body density inside —i.e., the gravity inversion problem—and for the body shape $\partial V_B$ itself. This stands in contrast to past approaches, for example, on gravity inversion—either employing a mascon perspective[44] or working in the mass density space directly[42,50,51]—which rely on the existence of a shape model for the body.

There are also notable similarities with other approaches. Previous approaches assume a kernel and create a highly parametric model of the body density inside a known volume and then fit it to reproduce gravitational observations[42]. This is to some degree similar to our approach, where the kernel functions are, analogously, defined by the neurons' non-linearities and the network weights are the model parameters. Stochastic gradient descent then allows fitting the model parameters to the observations. From this viewpoint, the main difference between past gravity inversion methods and the use of a geodesyNet stems from the mathematical properties of the parametric representation employed.

Unlike previous approaches, we use a feedforward artificial neural network and thereby rely on its universal function approximator property (see ref. [36]). This property makes it

particularly well suited to describe sharp density discontinuities, such as those encountered when crossing $\partial V_B$ or across possible interfaces between density layers. A more traditional polynomial or spline[51] parametrization, while in principle able to capture these effects, would require far too many coefficients to capture this feature with similar precision. In consequence, such density representations are typically limited to a given, known volume within the body[42] and are unable to represent the whole geodetic properties of a given body. Overall, geodesyNets replicate the results from previous approaches while providing a more holistic solution which requires fewer assumptions.

**On-board utilization**. The accurate characterization of the spacecraft's orbital environment is a crucial requirement of missions to comets and asteroids[43]. The approach we introduce here offers a potential simplification to this critical mission phase if the training of a geodesyNet can be performed on-board and in real-time while the spacecraft performs its orbits around the target body. This might save mission resources by eliminating the need to collect visual information for, e.g., a three-dimensional reconstruction of the shape. Further, if performed on-board the shape information can be collected online and less data may have to be transmitted. We argue that such a possibility, while currently not fully developed, is likely to become available in the near future. The use of dedicated on-board hardware enabling advanced artificial intelligence approaches for space missions has been recently reviewed by ref. [52] who discusses radiation-hardened GPUs as well as FPGAs or hardware accelerators such as Myriad 2[53]. In our case, as more batches of data would become available to the on-board computer, these could be seamlessly exploited by continuously adjusting the network parameters with some update rule to gradually improve the neural density field stored in the on-board geodesyNet. The network would be able to continuously learn during various mission phases—also accounting for unforeseen deviations and anomalies in the incoming data—and eventually be sent back to the ground as a compressed, differentiable, representation of the body shape and its first plausible internal structure. The memory requirement during training is perhaps the main limiting factor for the precision achievable in a possible on-board utilization, being mainly driven by the number of points $N$ used for the numerical quadrature used to evaluate the volume integral that defines the acceleration from a neural density field. Adjustments to enable a smaller batch size or a more memory-efficient numerical integration scheme may alleviate these concerns, however. On the other hand, the number of model parameters—supplementary methods, Table S4—is relatively modest (below 100,000 double precision parameters) and at most, a few hundreds of kilobytes are needed to store a trained geodesyNet. Note that reasonable accuracy for a rough estimate of the density distribution and induced acceleration is obtained even when using a small number of parameters.

## Methods
The training process shown in Fig. 1 is merely an outline and a number of important details were found to be important in training a geodesyNet. Hence, we describe the training setup in more detail and focus in even more depth on the choice of the loss function, the numerical integration method, and the differential training.

**Training setup**. At its core, the chosen neural network architecture is reminiscent of a SIREN network, as proposed by ref. [35]. It is a fully connected network with nine hidden layers of 100 neurons with sinusoidal activation functions in between layers. The final activation layer computes an absolute value (densities should be positive) or—in case of the differential training—a hyperbolic tangent (densities variations can be signed). A detailed analysis of the impact of the architecture details (SIREN's $\omega$, layers, neurons) is given in the supplementary methods, Tables S1, S2, and S5.

The network is trained using the ADAM optimizer with an initial learning rate of $10^{-4}$. The loss function has been specifically designed for geodesyNets and is presented in the supplementary methods, Supplementary Method 1. During training, a learning rate decay is used when the training loss plateaus for 200 iterations with a reduction with a factor of 0.8 up to a minimum learning rate of $10^{-6}$. Anecdotally, we observed several reductions in most training runs. The batch size during training is comparatively high at 1000 to average out the impact of noise in the form of numerical errors during the numerical integration. As a larger number of parameter studies is part of this work requiring hundreds of GPU hours, only one random seed was investigated for each run. Training consists of up to 10,000 iterations with early stopping on the training loss after a warmup of 3000 iterations, if 2000 iterations without a new optimum are computed. One iteration encompasses computing the ground-truth and model accelerations on one batch (1000) of points using a numerical integration sampling 500,000 points in the density field. Thus, the acceleration is computed at 10e8 points evaluating the neural density 5e10 times. The model with optimal training loss is then used for validation.

The synthetic observations which serve as the network's training data require a choice of sampling points at which the ground-truth acceleration is computed. The choice is critical to ensure numerical stability and plausibility (in practice, only sampling outside the body is plausible for spacecraft) while optimally providing information to the network. Hence, sampled points were sampled uniformly inside a unit sphere around the center of the body but always outside the body (the coordinate frame is chosen so that the body fits precisely inside $[-0.8, 0.8]^3$). The location of the sampled points (inside/outside the body) is determined with a custom PyTorch implementation of the ray-triangle intersection algorithm by Möller & Trumbore[54]. A low-poly version of the body meshes is used for the location determination. In a practical application, the sampled points would not be random, but be determined by, e.g., a spacecraft's trajectory or observed particles' locations. For computational efficiency, we resample new observations only every ten training iterations. Numerical integration and a mascon model then provide the ground-truth label as described in the previous sections. The network's prediction is analogously computed using the numerical integration over the neural density field. The method for the numerical integration is explored in more detail in the following sections.

To compute robust results on previously unobserved data we implemented a high-accuracy validation procedure. The validation consists of computing all reported error values on, respectively, 10,000 randomly sampled points at three specific altitudes above the body (0.04, 0.08, 0.2 length units). The sampling is displayed in supplementary methods, Fig. S3. The validation utilized a larger number (500,000) of samples in the numerical integration and high-fidelity - not low-poly - meshes for the location (inside/outside) determination.

## Data availability

All data and collected results needed to evaluate the conclusions in the paper is released online at https://zenodo.org/record/4749715#.YJrR6OhfiUk. The data were available under a Creative Commons Attribution 4.0 International license.

## Code availability

All code used to produce the results is available online at https://github.com/darioizzo/geodesynets. It is licensed under a GPL-3.0 open-source license.

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

## Acknowledgements

The authors are grateful to Dr. Francesco Biscani from the Max Planck Institute of Astronomy (Heidelberg) for making stable configurations of plausible planetoids available to be used as mascon ground truths in the paper and to Dr. Dawa Derksen for the interesting discussions and exchanges on neural scene representations.

## Author contributions

D.I. formulated and led the project, developed and implemented the theoretical calculations for the spherical harmonics and mascon gravity ground truths, and developed the masconCUBE approach. D.I. and P.G. refined the geodesyNET methodology and developed the code base, performed the numerical experiments, and wrote and revised the paper. P.G. automated the numerical experiments.

## Competing interests

The authors declare no competing interests.
