## [Peer Review File · Communications Engineering]

Geodesy of irregular small bodies via neural density fields:
geodesyNetsThis manuscript has been previously reviewed at another Nature Portfolio journal. This document only contains reviewer comments and rebuttal letters for versions considered at Communications Engineering.

Reviewers' comments:

Reviewer #1 (Remarks to the Author):

I still like the manuscript in general, but I still need clarification on some items. I have some questions regarding their answers in rebuttal. The main manuscript can be edited accordingly.

Rebuttal #1, "The gravity potential and acceleration is then computed from the density field evaluating the corresponding triple integral.":

How do you compute the triple integral? Is it just like how you do with mascons (i.e., discretization) or some blackbox method of your ML?

How do you define the locations of the density value you estimate? Is that automatically handled by ML during gravity inversion?

What does your density look like? Is it discrete value at each location or some kind of continuous function/interpolation? From the manuscript I have no idea what is actually estimated and how ML computes potential/acceleration. For example, if I have a prescribed density distribution (e.g., density as a function of spherical coordinates, polynomial, etc), how do I assign that to your ML? Also, is that triple integral exact or only approximation?

I have zero experience with ML but lots of experience with gravity estimation/inversion (e.g., Dawn, OREx, etc). Please address this paper to people like myself too so we have something to learn. The paper overall reads well, but it's a shame if this paper doesn't get the appreciation it deserves due to miscommunication.

Rebuttal #3, "The parameters of the network are the weights and biases which represent the density as a continuous field":

Ok, this is important. You have to mention this. Otherwise people like myself, who understand (conventional) gravity estimation/inversion but lacks ML knowledge, would be completely lost. So I ask again, how do you compute the nominal values for potential/acceleration? Once you have the density distribution, aren't you discretizing (basically mascon) and adding each contribution to potential/acceleration? What do you mean by numerically integrating? All methods nowadays are done by numerical integration. No one does this kind of computation by hand. And the same question from earlier still stands. If I already have a prescribed density distribution, how do I convert them into biases and weights?

Rebuttal #5, "The partials to compute the STM you refer to look indeed simple as they involve (trivially) the network derivatives with respect to the inputs, which are very efficient to compute": In the OD filter, we need the partials of spacecraft position/velocity w.r.t. ML parameters (which I think you said were density values and their coordinates but they really are weights and biases, right?). To compute that, we need the partials of the spacecraft acceleration w.r.t. ML parameters. That sounds like a lot of parameters. Is it really efficient? For each density point you estimate, you add an extra four parameters to estimate (coordinates + density value, or n weights and m biases), so your state vector grows very fast and your information/covariance matrix will likely suffer from high condition number. I understand this isn't the main theme of the paper though.

Reviewer #2 (Remarks to the Author):

This review will focus on the machine learning aspects of the presented method, as I have no knowledge of the field of gravitational inversion or of reported baselines.

I reviewed this paper previously for Nature Communications. The rebuttal have addressed all my concerns. I have no new concerns. Therefore I recommend this paper for publication.

Reviewer #3 (Remarks to the Author):

The revisions address all my comments on the previous submission to my satisfaction. I recommend that the article is accepted after correcting the minor typos listed below:

- Page 4, line 84: "both, body shape and density distribution, accurately" should be "both body shape and density distribution accurately"
- Page 4, line 86: "out" should be "our"
- Page 7, Line 149: two periods at end of blue text.
- Page 15, Line 309: I believe "specially" should be either "especially" or "specifically"
- Page 19, Lines 401-403: "The loss function... is presented in the next section." I believe this presentation is actually in the Supplementary Material
- Supplementary Material, Page 5: Still one case of "Russel" which should be "Russell"

Sean McArdle

Rebuttal for the reviews on our paper on geodesyNETs / Neural Density Fields

October 3, 2022

Reviewer 1

I still like the manuscript in general, but I still need clarification on some items. I have some questions regarding their answers in rebuttal. The main manuscript can be edited accordingly.

Rebuttal 1

“The gravity potential and acceleration is then computed from the density field evaluating the corresponding triple integral.”:

1. How do you compute the triple integral? Is it just like how you do with mascons (i.e., discretization) or some blackbox method of your ML?

Answer: The triple integral is computed by standard numerical integration techniques (a.k.a. numerical quadrature) from the expression (in the case of potential) $\int_V \frac{\rho}{r} dV$ where ρ is the output of our geodesyNET. The whole section “Numerical Integration” in the supplementary material is devoted to a detailed explanation on how this is done. Figure S1 contains even a study on the precision we get using different quadrature techniques to evaluate the triple integral. We have slightly amended the text to clarify this further.

2. How do you define the locations of the density value you estimate? Is that automatically handled by ML during gravity inversion? What does your density look like? Is it discrete value at each location or some kind of continuous function/interpolation? From the manuscript I have no idea what is actually estimated and how ML computes potential/acceleration.

Answer: The locations for sampling the density are chosen by the respective numerical integration scheme (e.g. Newton-Cotes, Monte Carlo Integration) when computing accelerations. We estimate a continuous function/interpolation (using your terminology) with the geodesyNET, so at the end of training the density is estimated as $(x, y, z) \mapsto \rho$. It looks like a continuous function, which we then plot in several figures (e.g. Figure 5, Figure 3) and we also provide animation for the 3D visualization of the density in the supplementary material. In the manuscript we explicitly explain: “At the end of its training, our network provides a differentiable expression mapping the position within a cubic volume V to a body density $\rho(x, y, z)$ compatible with the observed accelerations.” (differentiability also implies continuity). We also write “The numerical quadrature necessitates evaluation of the network at N distinct points inside the hypercube V . The quadrature is used to derive the gravitational acceleration from the density”, which answer to your question. We have slightly amended the text to clarify this further.

3. For example, if I have a prescribed density distribution (e.g., density as a function of spherical coordinates, polynomial, etc), how do I assign that to your ML? Also, is that triple integral exact or only approximation?

Answer: While this is not what our paper suggests nor shows, it is possible to train the network parameters (assign to our ML) a prescribed density by performing a simple regression directly over the network output (i.e. ρ). Note though, that the originality of our paper stems from the fact our ML does not find a density $\rho(x, y, z)$ from a known prescribed density distribution but directly from gravitational measurements without making any prior assumptions on the density distribution.

4. I have zero experience with ML but lots of experience with gravity estimation/inversion (e.g., Dawn, OREx, etc). Please address this paper to people like myself too so we have something to learn.

The paper overall reads well, but it's a shame if this paper doesn't get the appreciation it deserves due to miscommunication.

Answer: Indeed we are a bit caught in the middle of ML researchers and aerospace engineer practitioners and, believe us, it is difficult to target both communities, nevertheless we are trying. We simplified the ML part and we are glad other reviewers (experts in ML) still found it acceptable. We are at the European Space Agency and worked on HERA and we are in contact with the OPS team at ESOC in Darmstadt who are also already using some of our software (developed for different applications). We are aware that the current paper is only a first step and cannot go all the way to implement new procedures for immediate use in operational scenarios. We do intend to make the effort on the implementation side in the coming months/years after this first contribution is accepted and published. If this reviewer is interested in being involved in such an effort, please do contact us, we are welcoming discussions with other experts from the field.

Rebuttal 3

1. "The parameters of the network are the weights and biases which represent the density as a continuous field": Ok, this is important. You have to mention this. Otherwise people like myself, who understand (conventional) gravity estimation/inversion but lacks ML knowledge, would be completely lost. So I ask again, how do you compute the nominal values for potential/acceleration? Once you have the density distribution, aren't you discretizing (basically mascon) and adding each contribution to potential/acceleration? What do you mean by numerically integrating? All methods nowadays are done by numerical integration. No one does this kind of computation by hand.

Answer: We now mention this immediately after the equivalent sentence "A geodesyNET represents the body density directly as a function of Cartesian coordinates". To answer the follow-up question, we refer the reviewer to the paper text directly: "The network output, i.e. the neural density field, is then integrated numerically over the volume V . The numerical quadrature necessitates evaluation of the network at N distinct points inside the hypercube V ". We have now completed it with "The quadrature computes the gravitational acceleration at any point x, y, z resulting from the estimated density field ρ ". To further clarify, numerical integration can indeed be seen as equivalent to discretizing and adding each contribution to the potential/acceleration as you mentioned, for that reason we made a comparison to an equivalent mascon model and reported our findings in the supplementary material, section "A theoretical comparison between a mascon cube and a geodesyNet").

2. And the same question from earlier still stands. If I already have a prescribed density distribution, how do I convert them into biases and weights?

Answer: we hope that at this point this question has been clarified, nevertheless let us try a further explanation. While we do not propose nor study how to train (estimate) the network parameters (weights and biases) to match a prescribed density function (which we find a less interesting application), this is possible (and much easier) by just performing the "usual" training pipeline backpropagating (fitting to zero) the errors of the density predictions. Note, once again, that we do not do this, if this reviewer thinks that there is interest in doing so, please contact us as its a rather straight forward exercise.

Rebuttal 5

1. "The partials to compute the STM you refer to look indeed simple as they involve (trivially) the network derivatives with respect to the inputs, which are very efficient to compute": In the OD filter, we need the partials of spacecraft position/velocity w.r.t. ML parameters (which I think you said were density values and their coordinates but they really are weights and biases, right?). To compute that, we need the partials of the spacecraft acceleration w.r.t ML parameters. That sounds like a lot of parameters. Is it really efficient?

Answer: Yes. The cost of computing all those partial (independent from the number, even (say) 1,000,000 equals the cost of performing one single computation of the gravity/potential value at the point. It is the "miracle" of the reverse order numerical differentiation. Note that in forward mode automated differentiation, as well as in numerical differentiation, (which you maybe more familiar with?) this would not be true and each partial would cost as much as one single computation of the gravity/potential value. You are right, though, that to get a detailed description of the gravity field one may need a lot of parameters while for a NAV model one may want to limit the number to just a few (especially if we keep the spacecraft safely away from the irregular body). We think geodesyNETs are still of interest also in these simpler setups (Table S4 shows the error when the network dimension shrinks to only 600 params, you can see that at high altitudes - 0.25 the body radius - this is already rather small, but we have not

studied in detail a trade off with SH in these cases). I personally suspect that SH (since they converge nicely and quickly when far from the Brillouin sphere) can be better when the number of parameters estimated shrinks to only a few). On the other hand, if one wants to land safely or get close an irregular body, we may need to build a more accurate NAV model, in which case, whatever the method, more parameters will need to be estimated and geodesyNETs may prove to be a much better choice.

2. For each density point you estimate, you add an extra four parameters to estimate (coordinates + density value, or n weights and m biases), so your state vector grows very fast and your information/covariance matrix will likely suffer from high condition number. I understand this isn't the main theme of the paper though.

Answer: We do not think geodesyNETs would scale as you envision above during an OD scenario, but as mentioned we still have not worked on quantitative results for that particular scenario, so we cannot add much. Let us renew the option to contact us in case you are interested in discussing this research avenue.

Reviewer 2

This review will focus on the machine learning aspects of the presented method, as I have no knowledge of the field of gravitational inversion or of reported baselines.

I reviewed this paper previously for Nature Communications. The rebuttal have addressed all my concerns. I have no new concerns. Therefore I recommend this paper for publication.

Answer: Thanks for your time and for your effort.

Reviewer 3

The revisions address all my comments on the previous submission to my satisfaction. I recommend that the article is accepted after correcting the minor typos listed below:

- Page 4, line 84: "both, body shape and density distribution, accurately" should be "both body shape and density distribution accurately" - DONE

- Page 4, line 86: "out" should be "our" - DONE

- Page 7, Line 149: two periods at end of blue text. - DONE

- Page 15, Line 309: I believe "specially" should be either "especially" or "specifically"- DONE

- Page 19, Lines 401-403: "The loss function... is presented in the next section." I believe this presentation is actually in the Supplementary Material - DONE

- Supplementary Material, Page 5: Still one case of "Russel" which should be "Russell" - DONE

Answer: All implemented. Thanks for your time and for your effort.

REVIEWERS' COMMENTS:

Reviewer #1 (Remarks to the Author):

I have nothing to add/ask. Thanks for being so patient with all my questions!